# Task-related activity in human visual cortex

Zvi N. Roth *, Minyoung Ryoo, Elisha P. Merriam

Laboratory of Brain and Cognition, National Institute of Mental Health, NIH, Bethesda, Maryland, United States of America

* zvi.roth@nih.gov

## Abstract

The brain exhibits widespread endogenous responses in the absence of visual stimuli, even at the earliest stages of visual cortical processing. Such responses have been studied in monkeys using optical imaging with a limited field of view over visual cortex. Here, we used functional MRI (fMRI) in human participants to study the link between arousal and endogenous responses in visual cortex. The response that we observed was tightly entrained to task timing, was spatially extensive, and was independent of visual stimulation. We found that this response follows dynamics similar to that of pupil size and heart rate, suggesting that task-related activity is related to arousal. Finally, we found that higher reward increased response amplitude while decreasing its trial-to-trial variability (i.e., the noise). Computational simulations suggest that increased temporal precision underlies both of these observations. Our findings are consistent with optical imaging studies in monkeys and support the notion that arousal increases precision of neural activity.

**Data Availability Statement:** We make publicly available all of the fMRI data reported here, as well as Matlab code for analyzing the fMRI and eye-tracking data, and implementing the simulation (Roth & Merriam, 2020), at https://doi.org/10.17605/osf.io/cbjq6.

## Introduction

More than a decade ago, intrinsic signal optical imaging studies in awake behaving macaques identified a component of the hemodynamic responses in visual cortex that is tightly entrained to task timing, anticipates trial onsets, but is independent of visual stimulation [1]. This "task-related" response appeared to be spatially extensive (i.e., "global") because it extended throughout the imaged field of view. These results were surprising and controversial because this task-related hemodynamic activity does not correspond to simultaneously obtained electrophysiological measurements from visual cortex [1], calling into question the link between hemodynamics and neural activity. Moreover, it is not clear whether or not such activity is even functionally relevant for task performance, raising the possibility that task-related hemodynamic activity is simply a measurement artifact or worse, an irrelevant epiphenomenon.

Multiple lines of evidence suggest that global functional MRI (fMRI) activity is not related to neural computation. Several studies have demonstrated hemodynamic, vascular, or motion-related artifacts that are "global" and hence resemble task-related activity [2–7]. Other physiological covariates, such as changes in heart rate, blood pressure, and respiration, have been shown to correlate with global brain responses [8–10]. This view is so prevalent in the field that a large number of studies estimate and then analytically remove global fMRI activity as a preprocessing step to improve the signal-to-noise ratio of fMRI time-series measurements [11, 12].

**Funding:** This research was supported by the Intramural Research Program of the NIH (ZIA-MH002909), under National Institute of Mental Health Clinical Study Protocol 93-M-1070 (NCT00001360). The funders had no role in study design, data collection and analysis, decision to publish, or preparation of the manuscript.

**Competing interests:** The authors have declared that no competing interests exist.

**Abbreviations:** BOLD, blood oxygen level–dependent; EVC, early visual cortex; fMRI, functional MRI; HRF, hemodynamic response function; IRF, impulse response function; LFP, local field potential; ME-ICA, multi-echo independent components analysis; ROI, region of interest; std, standard deviation.

An alternative possibility, however, is that global fMRI activity corresponds to changes in arousal, which continually fluctuates as participants perform a task [13, 14]. There is increasing appreciation of the relationship between arousal, pupil size, and cognitive performance [15–19]. For example, unlike spatial attention, which increases perceptual sensitivity in a local region in visual space, changes in arousal lead to pupil dilation and do not necessarily increase sensitivity [15, 20, 21]. If this hypothesis is correct, then task-related hemodynamic activity may contain important information about cognitive processes, which is overlooked by the vast majority of fMRI experiments.

We tested the hypothesis that task-related activity tracks arousal. We used a task in which we could manipulate brain state, without changing the sensory or motor requirements of the task. This experiment was designed specifically to measure task-related activity in visual cortex using a protocol and analysis procedure analogous to that used in earlier studies in nonhuman primates [1, 22–24]. We used a simple, periodic task combined with a performance-contingent reward protocol that has been used extensively to alter participants' motivation and arousal [25]. If the task-related response tracks changes in arousal, it would indicate the functional relevance of this response and suggest a direct link to cognitive processes.

## Results

We aimed to isolate and characterize task-related fMRI activity in humans and to test the relationship between this activity and arousal, which we manipulated with reward. Participants performed a periodic orientation discrimination task and gained money based on their performance (Fig 1A). On high-reward runs, participants could gain a relatively large sum of money. On low-reward runs, they could gain much less. Eye tracking performed outside the scanner indicated that participants maintained central fixation and did not make saccades toward the stimulus following its appearance (S3 Fig). Saccades followed the main sequence (S3A Fig), were small in amplitude relative to the stimulus eccentricity (S3B and S3C Fig), and showed a typical horizontal bias (S3D Fig) [26].

### Task-related activity in human visual cortex

We observed widespread fMRI activity that extended throughout much of the imaged field of view, covering all of occipital cortex in both hemispheres, as well as posterior temporal and parietal cortex (Fig 1B). Two observations suggest that this activity is of similar origin to the optical imaging measurements in macaque [1]. First, this activity was entrained to task timing with a similar periodicity. Second, this activity, although present in visual cortex, was not related to the visual stimulus. The stimulus was a small, brief, peripheral grating that was expected to evoke activity primarily in the corresponding retinotopic location in the contralateral hemisphere. The behavioral protocol involved spatial attention directed to the stimulus, and spatial attention was also expected to evoke activity in the corresponding retinotopic location in the contralateral visual cortex. Hence, the widespread activity that we observed in early visual cortex (EVC) of the ipsilateral (i.e., right) hemisphere is not likely attributable to feedforward stimulus drive or to spatial attention.

To directly test whether the stimulus itself contributed to the activity that we measured, we performed two additional analyses. First, in a separate experiment, participants performed a periodic button-press task that had the same timing as the task in the main experiment but did not involve a peripheral visual stimulus. We observed the same widespread task-related response that extended through visual cortex, extending from the foveal to the most peripheral representation (S1 Fig). Second, estimates of the receptive field location for each voxel were

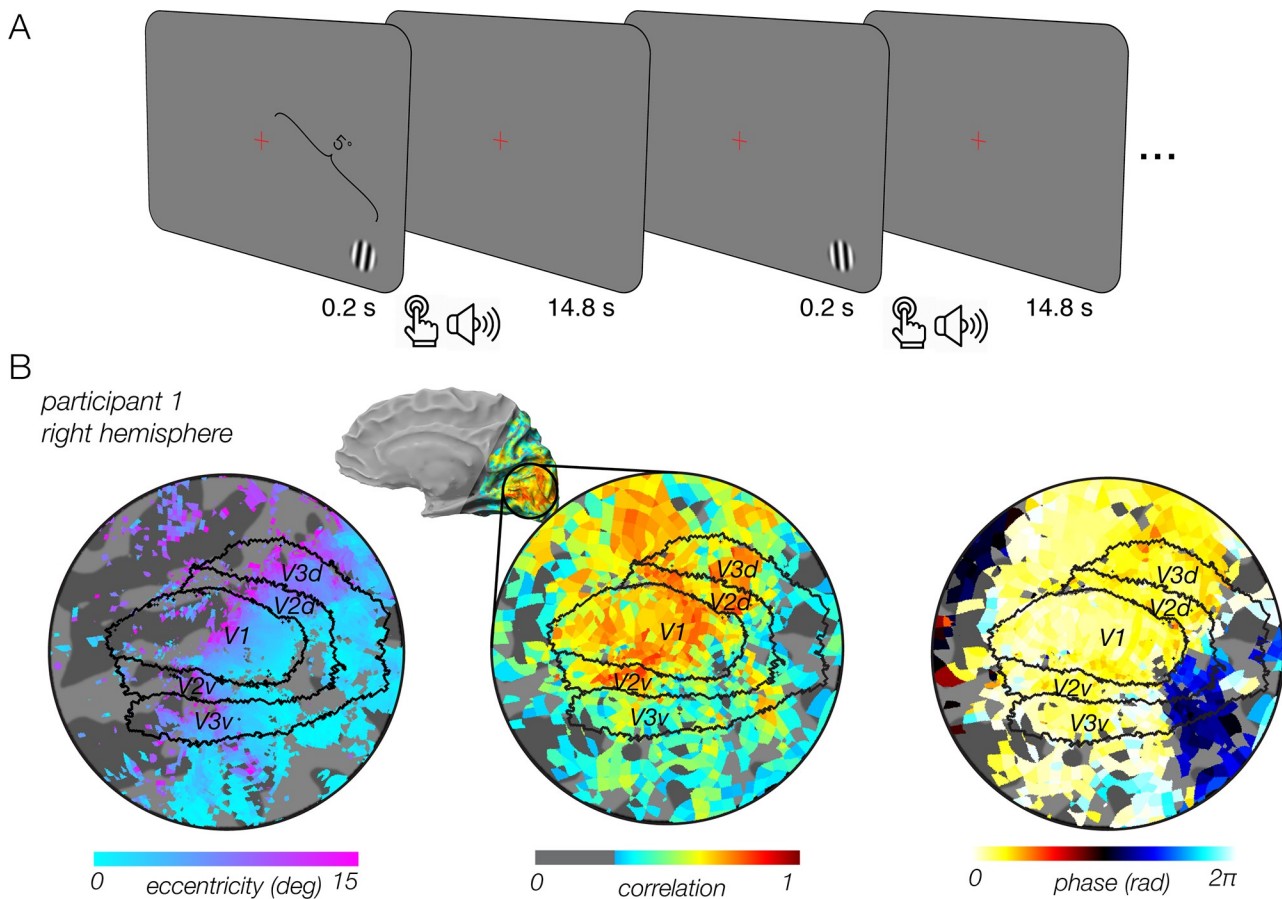

**Fig 1. Experiment design and task-related fMRI activity in early visual cortex.** (A) Experiment design. Participants were instructed to continuously fixate on a central cross while performing a 2AFC orientation discrimination task on a peripheral stimulus. On each trial, a small grating was briefly presented for 200 ms in the bottom right of the screen. Participants indicated whether it was tilted CW or CCW relative to vertical and received immediate auditory feedback. Participants maintained fixation until the next trial. In each run, participants could gain either a high or low monetary reward for correct performance. (B) Medial view (inset), and a flattened map of ipsilateral visual cortex (bottom panels) of participant P1. Left: visual eccentricity. Hue indicates eccentricity of the population receptive field center for each voxel. Retinotopic borders of V1–V3 were defined by an anatomical template extending to 80˚ eccentricity, well beyond the spatial extent of the screen. Map threshold, r² > 0.1. Shaded region on medial views indicates cortex not included in the imaged field of view. Center: response correlation, showing a widespread fMRI response linked to task timing. Map threshold, r > 0.3. Hue indicates correlation with best-fitting cosine at the task frequency. Right: response phase. Same threshold as middle panel, with hue indicating phase of best-fitting cosine for each voxel. Phase values indicate the response latency for each voxel. Underlying data can be found at https://osf.io/cbjq6/. 2AFC, two-alternative forced choice; CCW, counterclockwise; CW, clockwise; fMRI, functional MRI.

used to create visual field coverage plots from the responses in EVC. Visual field plots for the main experiment were far more spatially extensive than for the stimulus localizer (S2 Fig). These observations suggest that the task-related response was not related to the visual stimulus in the main experiment.

Task-related responses exhibited two distinct temporal profiles. Throughout most of visual cortex, voxels exhibited a peak response at approximately 6 s (yellow-white voxels), indicating that these responses were linked to the onset of the trial, taking into account the temporal dynamics of the blood oxygen level–dependent (BOLD) fMRI response (Fig 1B, right). A second subset of voxels, located near the fovea, peaked later in the trial (evidenced by blue-black hue). These later responses at the fovea suggest that the task-related response may reflect multiple cognitive operations that vary with visual eccentricity.

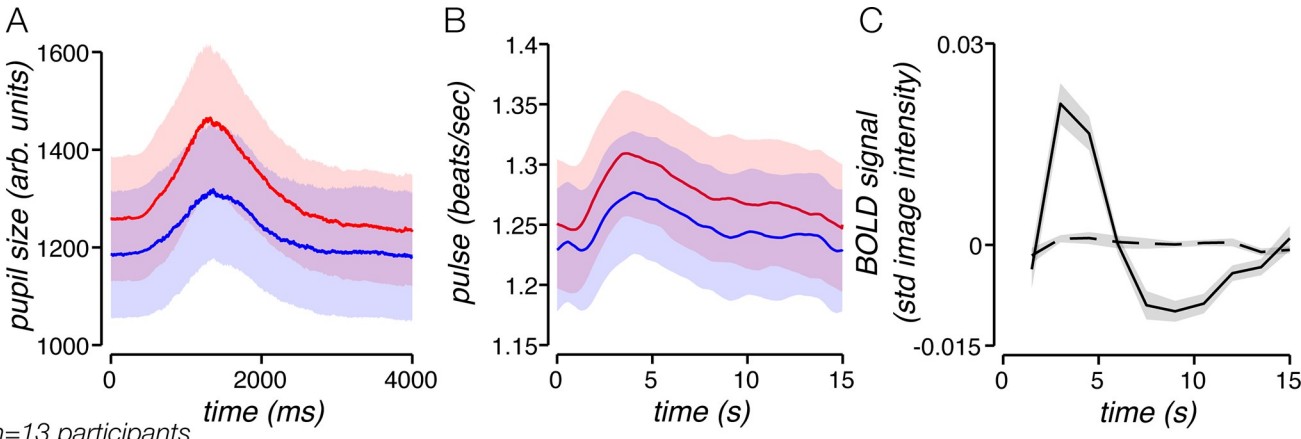

**Fig 2. Reward modulates arousal, evident in pupil size and heart rate.** (A) Mean pupil size for high- and low-reward trials. Pupil size exhibited a response that was time-locked to trial timing, showing an increase at the beginning of the trial followed by a return to baseline by 4 s. High-reward runs (red) evoked larger task-related pupil changes than low-reward runs (blue). (B) Heart rate for mean trial, averaged across participants. Heart rate exhibited a task-related response and was greater for high reward (red) than for low reward (blue). (C) Pulse-to-BOLD kernel before (solid line) and after (dashed line) global signal regression. Shaded regions, ±SEM across participants. Underlying data can be found at https://osf.io/cbjq6/. arb., arbitrary; BOLD, blood oxygen level–dependent; std, standard deviation.

## Reward affects arousal: Behavior, pupil size, and physiology

We observed changes in both pupil size and heart rate with reward level, in the absence of changes in perceptual performance, suggesting that reward affected participants' level of arousal, rather than the allocation of spatial attention.

We observed a periodic change in pupil size that was entrained to the timing of the task (Fig 2A). This change in pupil size was likely not associated with a pupillary light reflex to the appearance of the stimulus, since the grating stimulus had the same mean luminance as the gray background. Instead, we attribute the modulation in pupil size to cognitive processes related to the task [27, 28]. We identified two components of the pupil measurement on each trial: a phasic component, which was entrained to the timing of the trial, and a tonic component, defined as the baseline pupil size on that trial. We found that the amplitudes of both phasic and tonic pupil components were larger on high-reward trials than on low-reward trials ($p < 0.0001$, two-sided permutation test, for both tonic and phasic pupil size modulation). We infer from the modulation in pupil size that reward magnitude led to an increase in arousal, in line with previous studies that used pupil size to infer changes in arousal [29–31].

We monitored heart rate during task performance, since changes in heart rate are thought to be an indication of arousal state [32]. Reward level affected heart rate in two ways. First, mean heart rate was elevated during high-reward runs relative to low-reward runs ($p < 0.0001$, permutation test) (Fig 2B). This may reflect a tonic change with arousal, similar to the baseline shift that we observed for pupil size on high-reward runs (Fig 2A). Second, we found that heart rate exhibited a task-related response, increasing slightly after the onset of the stimulus (Fig 2B). This task-related heart-rate change was itself modulated by reward ($p = 0.0012$, permutation test) (Fig 2B), much in the way that the phasic pupil size was modulated by reward. Together, the changes in heart rate that we observed, along with the modulation in pupil size, suggest that reward level was an effective means of modulating arousal level.

We wondered whether reward had an impact on behavioral performance during the scan. Reward did not affect accuracy on the orientation discrimination task (high reward, 82.7% ± 6.5%; low reward, 81.0% ± 7.8% [mean ± standard deviation (std)], $p = 0.36$, paired $t$ test) nor

on reaction time (reaction time: high reward, 631 ms ± 127 ms; low reward, 633 ms ± 99 ms [mean ± std], permutation test *p*-value: 0.39). We interpret these behavioral results to indicate that increased reward did not affect the allocation of spatial attention [20].

## Breaking the link between heart rate and BOLD

The goal of our analysis was to test the link between arousal and task-related fMRI activity. However, the changes in heart rate that we observed created an important obstacle for making inferences regarding fMRI activity, for the following reason: it is conceivable that arousal affects physiological processes, which in turn impact the BOLD signal. Although multi-echo independent components analysis (ME-ICA) considerably reduces physiological noise in the fMRI data [33, 34], it does not eliminate it [35]. Any change in fMRI activity with reward could, in theory, reflect peripheral physiological changes rather than neuronal changes. Indeed, we found that heart rate influenced the BOLD signal in a systematic way, yielding a pulse-to-BOLD kernel [36, 37] (Fig 2C). This observation suggests that reward influences heart rate, which in turn affected the BOLD signal.

Could the task-related response reflect modulations in physiological covariates rather than changes in the brain? To address this question, we removed the impact of physiological signals from the fMRI time series by regressing out the global mean fMRI time series, a procedure that is thought to be the most effective means of removing the impact of heart rate and respiration on fMRI measurements [4, 38]. This procedure reduced the mean pulse-to-BOLD kernel amplitude by 92% (Fig 2C), confirming global signal regression is effective at mitigating the impact of heart-rate effects in fMRI. Critically, task-related fMRI activity remained robust after regressing out the global signal (Fig 3), indicating that task-related activity is not a secondary consequence of the respiratory and pulse changes that occur with the task.

## Reward modulates task-related fMRI activity

To evaluate the relationship between reward and task-related activity, we averaged the fMRI response across voxels within three eccentricity bins roughly corresponding to the retinotopic location of the fixation cross (0–1 deg), the location of the visible screen (1–10 deg), and beyond the screen boundary extending to the most eccentric extent of V1 (10–80 deg). This

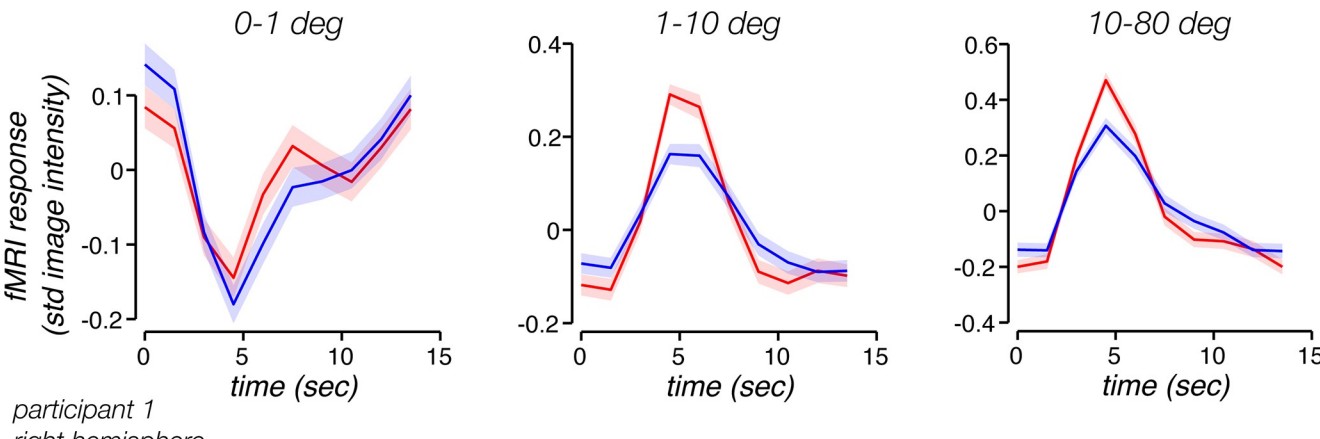

participant 1
right hemisphere

**Fig 3. Task-related response as a function of visual eccentricity.** fMRI responses in three subregions of EVC defined by visual eccentricities, for a representative participant (P1). Left, <1 deg eccentricity; center, 1–10 deg eccentricity; right, >10 deg eccentricity. Time series from EVC were averaged across all voxels within each bin and across all trials within high-reward (red) and low-reward (blue) runs. Shaded regions, ±SEM across trials. Underlying data can be found at https://osf.io/cbjq6/. EVC, early visual cortex; fMRI, functional MRI; std, standard deviation.

analysis supported three observations. First, we found that task-related activity was robust at all three eccentricity ranges (Fig 3). The fact that we observed strong responses from voxels well beyond the screen boundary confirms that task-related activity was not associated with a visual event on the screen. Second, the response of voxels at the foveal representation had a temporal profile distinct from voxels at other eccentricities, suggesting that activity at the foveal representation may reflect a distinct computation. Finally, we found a robust and reliable modulation in the response due to reward level (Fig 3, blue and red curves). Reward level had an impact on multiple measures of task-related activity. Responses were typically larger and less variable on high-reward runs. Here, we characterize these effects and show that they can best be modeled by changes in temporal precision.

We evaluated the impact of reward using two complementary measures of response amplitude: std and Fourier amplitude of the trial-averaged time series. Both measures revealed significantly greater task-related response amplitude for high reward than for low reward (STD measure, $p = 0.008$; Fourier amplitude measure, $p = 0.021$; two-sided permutation test for both). We quantified the latency of the task-related response by computing the Fourier phase of the trial-averaged time series. Activity latency during high-reward was slightly later than for low reward ($p = 0.041$, two-sided permutation test).

Foveal and peripheral EVC generally displayed responses at different latencies. Averaging the responses across voxels at different eccentricities could potentially obscure the impact of reward, as the different response profiles could cancel each other. To systematically study how the task-related response amplitude and latency covary with eccentricity, we divided EVC into 12 exponentially spaced eccentricity bins and analyzed the amplitude and phase of the response within each of them. The amplitude was high at the fovea but quickly dropped, reaching a trough at around 3 deg, before rebounding at higher eccentricities (Fig 4A). The phase of responses at the fovea differed from those at higher eccentricities by approximately 180˚, resembling an inverted response. The trough in response amplitude may reflect destructive interference that occurs when responses of opposite phase are averaged within an eccentricity bin. Although the task-related response was heterogeneous across eccentricities within EVC

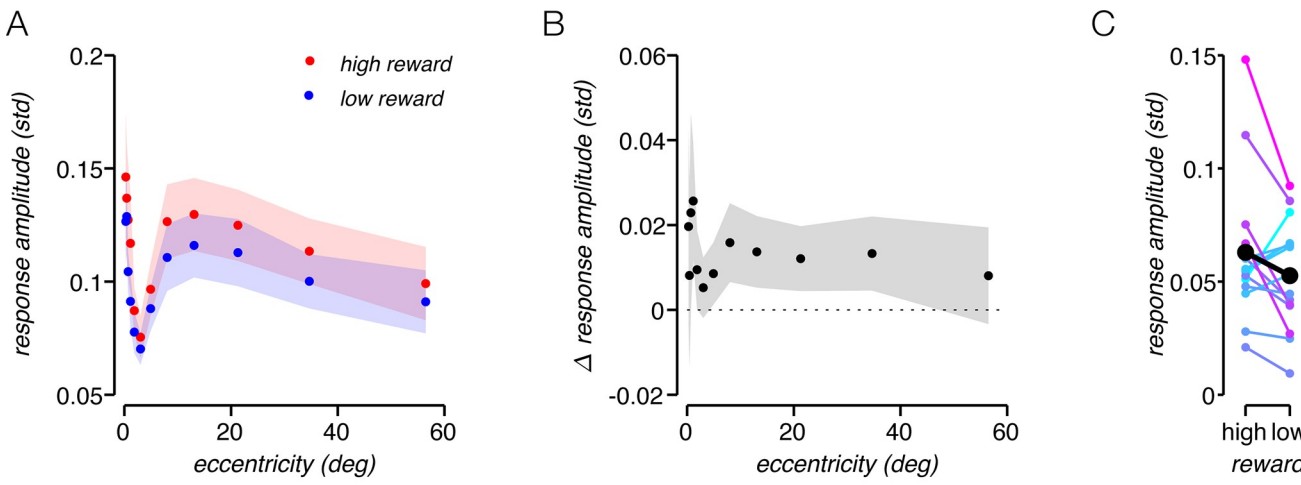

n=14 participants

**Fig 4. Task-related response amplitude is modulated by reward.** (A) EVC task-related response amplitude as function of eccentricity. Shaded regions, ±SEM across participants. (B) Low-reward amplitude subtracted from high-reward amplitude as function of eccentricity. (C) Amplitude of EVC task-related response, for all participants. Amplitude was quantified by the std of the average trial. Data points and lines connecting high- and low-reward amplitudes are colored according to the difference between high- and low-reward amplitude for each participant. Underlying data can be found at https://osf.io/cbjq6/. EVC, early visual cortex; fMRI, functional MRI; std, standard deviation.

(Fig 4A), response amplitude was consistently nominally higher for high-reward than for low-reward runs, across all eccentricities (Fig 4B). The consistent effect of reward on the task-related response amplitude indicates that the response is related to cognitive processes modulated by reward.

Both response amplitude and response latency were measured by averaging across trials in order to maximize the signal-to-noise ratio. However, such averaging could obscure subtle changes in the fMRI activity associated with reward. For example, stimulus onset, motor initiation, and attention have all been shown to decrease trial-to-trial neural variability [39, 40]. It is conceivable that reward magnitude lowers variability as well. To test this possibility, we measured the variability across trials for each time point in the fMRI response and computed the mean variability across time points, yielding a measurement of average time-point variability. We found that time-point variability was significantly lower for high- than for low-reward trials ($p = 0.001$, one-sided permutation test), for each eccentricity bin ($p < 0.05$ for all, one-sided permutation test) (Fig 5A).

Time-point variability is a general measure of noise in the signal and could encompass several distinct sources of variability, including sources of noise related to the task, as well as sources of noise from ongoing neural fluctuations and fMRI measurement noise. We performed two additional analyses to quantify components of variability directly associated with the task-related response itself. First, we computed the std of the amplitude of the task-related response across individual trials. Second, we computed the circular std of the phase, or timing, across trials. We found that variability in both amplitude and timing was lower during high-reward runs ($p = 0.042$ and $p = 0.02$, respectively, one-sided permutation test, Fig 5B and 5C). To conclude, we found that higher reward resulted not only in a higher task-related response amplitude but also in lower variability.

To test whether the effect of reward on fMRI activity was due to a modulation of stimulus-evoked responses, we repeated all analyses after excluding voxels that responded to the localizer at a coherence threshold of $r > 0.3$. This resulted in excluding an average number of voxels equal to 16.2% of the EVC region of interest (ROI; range 7.23%–31.55% across participants). We then repeated all analyses and found that excluding voxels did not alter any of our main findings. Specifically, after excluding visually responsive voxels, high reward significantly increased response amplitude ($p = 0.001$) and decreased time-point variability ($p = 0.007$) and phase variability ($p = 0.025$). However, we found that after excluding visually responsive voxels, amplitude variability did not decrease significantly with reward ($p = 0.14$). We conclude that effects of arousal reflect primarily changes in task-related activity and are not due to a modulation of stimulus-evoked activity.

## Reward increases temporal precision of task-related response

What could be driving the decrease in variability? We identified three distinct possibilities. First, on high-reward trials, there may be a decrease in ongoing neural fluctuations, which are independent of the task. Second, reward could attenuate fluctuations in the task-related response amplitude. Finally, a third possibility is that reward decreases trial-to-trial fluctuations in the latency of the task-related response.

To differentiate between these three possibilities, we implemented a simulation of fMRI activity that instantiated task-independent noise, amplitude jitter, and temporal jitter. All three noise sources may be present in the measured fMRI times series, but they are not necessarily all modulated by reward. We generated noiseless time series consisting of periodic task-related responses, of a fixed amplitude and latency (Fig 6A and 6E). We then added the three possible noise sources and measured the resulting simulated task-related response. Assuming

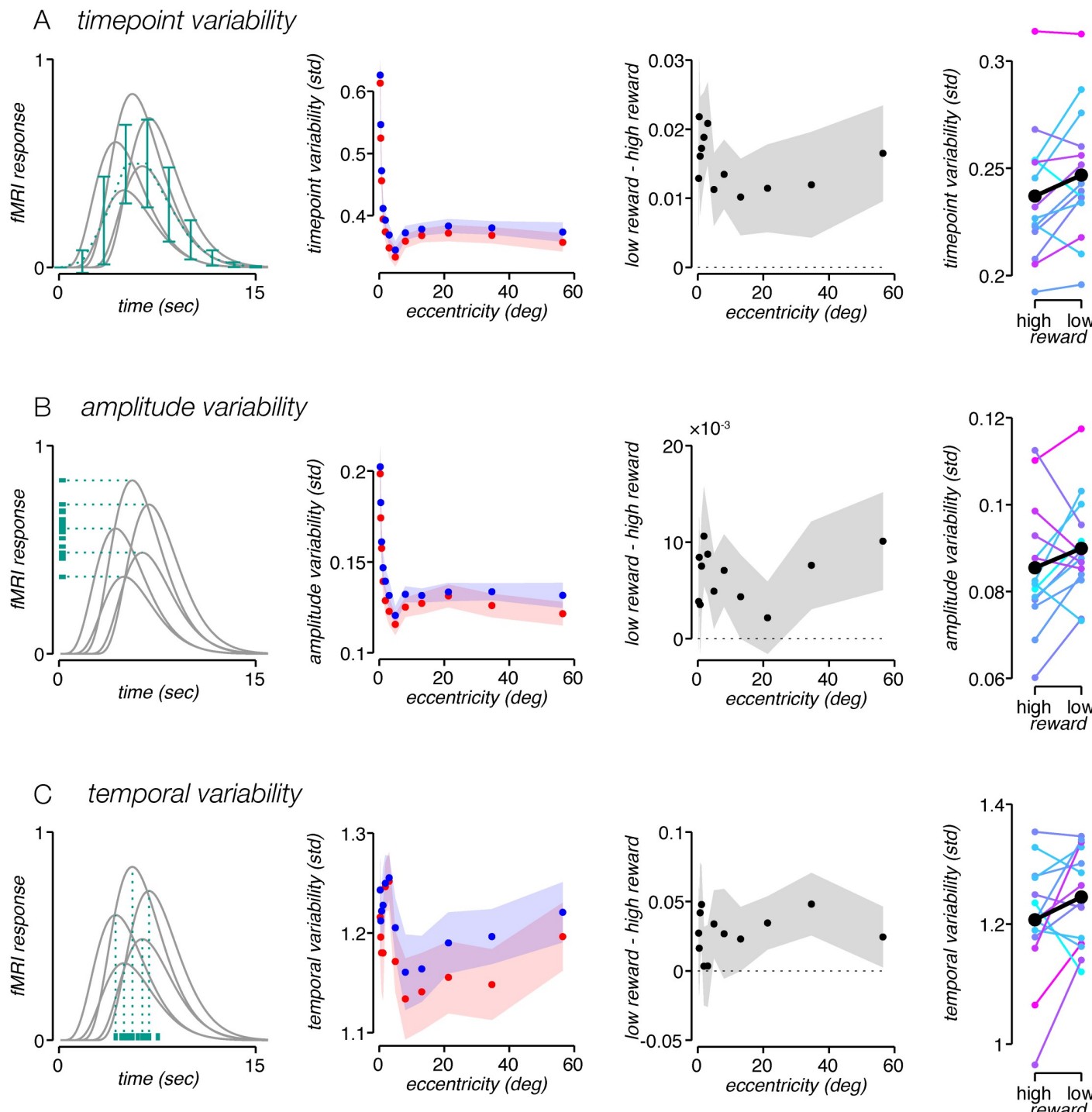

**Fig 5. Higher reward decreases three measures of response variability.** (A) Time-point variability of task-related response. Leftmost panel, schematic illustration of analysis. Gray lines, simulated task-related responses; green dotted line, mean response; green error bars, std at each time point. Time-point variability was quantified by taking the std of each time point in the task-related response, averaging across time points, and then averaging across participants. Second panel, time-point variability as function of eccentricity, for high (red) and low (blue) reward. Third panel, high-reward time-point variability subtracted from low-reward time-point variability as function of eccentricity. Shaded regions, ±SEM across participants. Rightmost panel, mean time-point variability of EVC task-related response, for all participants. Time-point variability was significantly lower for high- than for low-reward trials. Data points and lines connecting high- and low-reward variabilities in all right panels are colored, as in Fig 4C, according to the difference between high- and low-reward response amplitude for each participant. (B) Leftmost panel, schematic illustration of analysis. Green horizontal lines, amplitudes of simulated responses. Amplitude variability was quantified by first computing the amplitude (i.e., std) of each trial, then

computing the std across trials, and finally averaging across participants. Second panel, amplitude variability as function of eccentricity, for high (red) and low (blue) reward. Third panel, high-reward amplitude variability subtracted from low-reward amplitude variability as function of eccentricity. Shaded regions, ±SEM across participants. Rightmost panel, amplitude variability of EVC task-related response, for all participants. Amplitude variability was significantly lower for high- than for low-reward trials. (C) Leftmost panel, schematic illustration of analysis. Green vertical lines, latencies of simulated responses. Temporal variability was quantified by taking the circular std of the phase of the Fourier component corresponding to a single cycle per trial, and averaging across participants. Second panel, temporal variability as function of eccentricity, for high (red) and low (blue) reward. Third panel, high-reward temporal variability subtracted from low-reward temporal variability as function of eccentricity. Shaded regions, ±SEM across participants. Rightmost panel, temporal variability of EVC task-related response, for all participants. Temporal variability was significantly lower for high- than for low-reward trials. Underlying data can be found at https://osf.io/cbjq6/. fMRI, functional MRI; EVC, early visual cortex; std, standard deviation.

that reward reduces the amount of noise, we tested for each noise type whether noise reduction reproduces the effects reward has on task-related activity.

We found that all three noise sources cause increases in time-point variability, amplitude variability, and temporal variability. However, the noise sources each had a distinct impact on the task-related response. First, they differed in the shape of the time-point variability time series they produced. A change in the amount of noise unrelated to the task response affects the variability at all time points equally (Fig 6B and 6F). This is because the source of time-point variability is the noise that was added independently to each time point. Such an effect diverges from the empirical fMRI variability time series, which is not flat but rather follows a trajectory similar to the task-related response itself (Fig 7B). However, changes in both amplitude and temporal jitter can yield simulated time-point variability time series that are similar to the fMRI variability time series, depending on the shape of the simulated impulse response function (IRF) used (Fig 6C, 6D, 6G and 6H). Second, the sources of noise varied in how they affected the mean amplitude of the response. Whereas independent noise and amplitude jitter have no systematic effect on the mean amplitude, increasing temporal jitter lowers the mean response amplitude (Fig 6 and S6 Fig). Independent noise has no systematic effect on the mean amplitude, since the noise is independent of the response, and averaging across trials reduces any random impact on the amplitude (Fig 6B). Similarly, amplitude jitter involves random amplitude fluctuations, which again cancel out when enough trials are averaged (Fig 6C). However, temporal jitter does not have the same impact on amplitude after averaging trials. Greater temporal jitter translates to poorer temporal alignment of responses across trials, which only partially cancel each other out. Temporal jitter results in a wider response which is also smaller in amplitude (Fig 6D).

To sum up the simulation results, a decreased amount of temporal jitter reproduced effects of reward on task-related activity, indicating that modulation of temporal jitter by reward level can potentially be the source of changes in response amplitude, time-point variability, amplitude variability, and temporal variability. As such, it is the most parsimonious explanation. However, we cannot rule out a combination of temporal jitter and other sources of noise, such as amplitude jitter and noise unrelated to the task.

## Discussion

Here, we report spatially widespread fMRI activity in a simple perceptual task. This activity is not evoked by the stimulus, since there was never a stimulus contralateral to the hemisphere that we analyzed. Nor was this activity related to the global fMRI signal, often removed in pre-processing of resting-state fMRI experiments, since we regressed out the global signal before analyzing the data. Instead, it is likely that the response that we measured shares the same origin as a hemodynamic signal measured in optical imaging studies in monkeys [1] using similar tasks and behavioral protocols. We found that this widespread fMRI activity was modulated by reward, suggesting that it is functionally relevant. Similar reward-dependent modulations

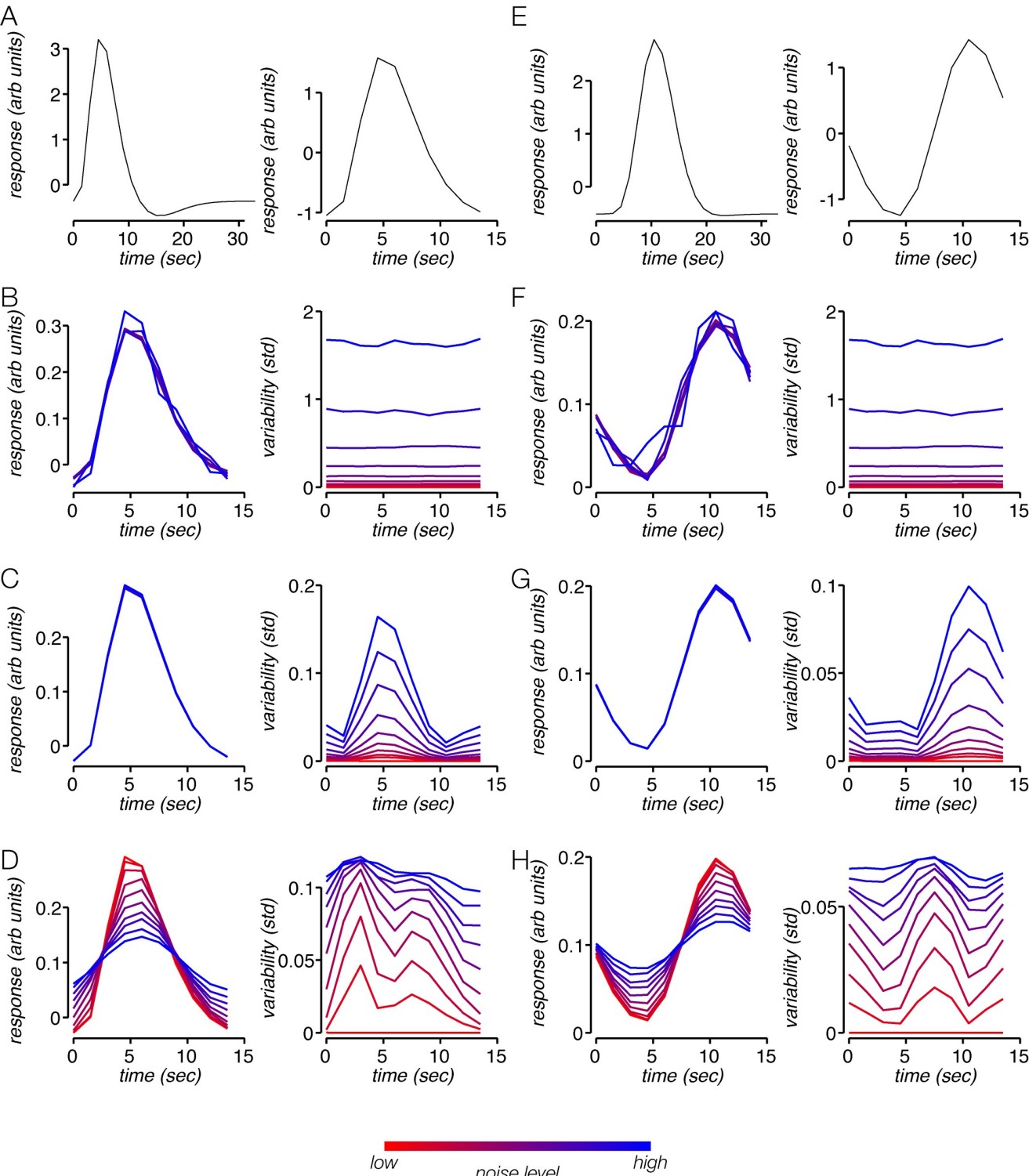

low　　noise level　　high

**Fig 6. Impact of three distinct noise sources on measures of hemodynamic variability.** (A) Left, IRF used for the first simulation. Right, average trial response with no noise. The response differs slightly from the IRF because the previous trial has a prolonged influence on the signal. (B) Left, average trials for independent noise ranging from minimal (red) to maximal (blue), for first simulation. Right, average time-point variability time course for the different independent noise levels. (C) Left, average trials for response amplitude jitter ranging from minimal (red) to maximal (blue). Right, average time-point variability for the different amplitude jitter levels. (D) Average trials for response temporal jitter ranging from minimal (red) to maximal (blue). Right, average time-point variability for the different temporal jitter levels. (E-H) Same as (A–D), for second simulation, using a different IRF. arb., arbitrary; IRF, impulse response function; std, standard deviation.

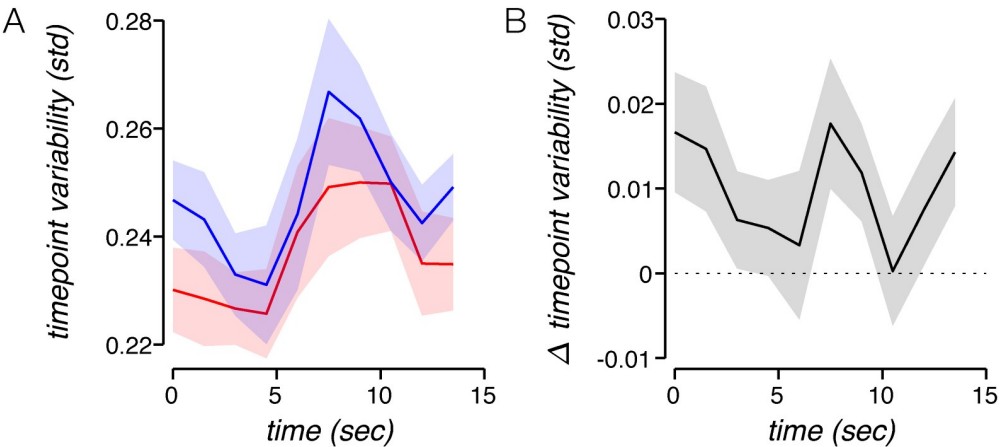

**Fig 7. Higher reward decreases time-point variability dynamically throughout the trial.** (A) Mean group time-point variability time series. At each time point, variability was computed per participant and then averaged. Time-point variability changes systematically throughout a trial, in a similar way for low-reward and high-reward runs. Shaded regions, ±SEM across participants. (B) High-reward time-point variability subtracted from low-reward time-point variability. Time-point variability was greater for low reward than for high reward across all time points, and the difference between the two exhibits a temporal profile similar to temporal variability for each of the two reward conditions. Shaded regions, ±SEM across participants. Underlying data can be found at https://osf.io/cbjq6/. std, standard deviation.

were also observed for pupil size and heart rate. Taken together, these results suggest that task-related fMRI activity is related to arousal, although the link between the two may not be causal.

The BOLD fMRI time series is a composite measurement, composed of multiple neural signals and sources of noise [2]. Although it is often assumed that "global" fMRI responses are noise [41], several observations indicate that task-related activity is signal, not noise. First, we took several steps to mitigate the influence of measurement noise. We collected fMRI time-series data using an acquisition strategy and analysis pipeline (ME-ICA) that is known to reduce noise and directly estimate BOLD effects [4, 33, 34]. We also removed from the data in preprocessing a brain-wide "global" signal, a procedure that is thought to remove sources of noise related to respiration [4, 38, 42]. Second, the modulation of the task-related response with reward resembled the modulation of pupil size. Since pupil size is widely regarded as a proxy for arousal [13, 43–45], it is likely that task-related activity is also related to arousal.

## Manipulating arousal with reward

Several lines of evidence suggest that the reward protocol that we used successfully manipulated participants' level of arousal. First, pupil size, a well-studied proxy for arousal [29, 46], was larger on high-reward runs, exhibiting both tonic and phasic changes with reward level. Second, heart rate, which has been shown to correspond to arousal and motivation [23, 32, 47], was higher during high-reward runs. Finally, arousal did not affect task performance or reaction time, consistent with earlier reports that reward increased arousal while not affecting the allocation of spatial attention, with the concomitant enhancement in perceptual sensitivity [20]. This would imply that even on low-reward runs participants performed the task as best as they could. The increase in reward, although providing additional incentive, did not provide any additional cognitive resources that could be allocated toward increasing perceptual sensitivity, and therefore, increased reward did not impact task performance. It is likely the case that other experimental manipulations (e.g., cold-water shock or other punishment) may have

similar effects on arousal levels. Punishment and reward may both increase arousal and, therefore, modulate task-related activity in a similar way.

## Relationship to previous studies

Several previous studies reported activity in V1 that was not evoked directly by visual stimuli [48–50]. In these studies, however, responses were measured from cortical subregions that corresponded retinotopically to where covert spatial attention was presumably allocated. Similarly, a series of studies observed widespread activity that was associated with task switching in addition to retinotopic stimulus- and cue-evoked activity in visual cortex [51–53]. However, the effects reported in these studies are categorically distinct from task-related activity, since they involve responses related either to visual stimuli or to spatial attention.

The task-related activity that we report here may correspond to a nonperceptual response reported by Jack and colleagues [54]. In that study, although the stimulus was bilateral, the response did not seem to correspond to the retinotopic location of the visual stimulus and was distinct from an attentional response that was measured in the same experiment [54]. Critically, however, the nonperceptual response reported by Jack and colleagues [54] was not modulated by task factors such as difficulty and pace, whereas activity that we measured was modulated by reward.

The task-related activity that we report may also be related to the global BOLD component observed during motion-induced blindness [55]. In that study, the global response was observed most prominently at eccentricities that were beyond the retinotopic location of the stimulus. The polarity of this component corresponded to perceptual appearance and disappearance of a target stimulus. Since properties of this component are similar to the task-related responses that we report here, future studies should investigate which perceptual properties are reflected in the task-related response.

Yellin and colleagues [56] measured pupil size during resting-state fMRI scans. After convolving the pupil size trace with a hemodynamic response function (HRF), they found that larger pupil size corresponded to lower BOLD signal across most of the cortex, including visual cortex. Similarly, Broday-Dvir and colleagues [57] found that on unattended trials without stimuli, higher pupil size corresponded to lower BOLD responses in visual cortex. Chang and colleagues [58] tracked whether monkeys' eyes were open or closed while measuring both the BOLD signal and local field potentials (LFPs), without a task, in darkness. They found that when monkeys closed their eyes, activity increased throughout cortex, including visual cortex. The pattern of activity reported by these studies is opposite to our finding that higher reward results in higher activity in visual cortex, and suggests that the relationship between arousal and activity may not be a monotonic function. Specifically, at moderate arousal levels, a slight increase in arousal may correspond to an increase in activity, and at very low arousal levels (e.g., when on the verge of sleep), an arousal increment may correspond to decrease in activity. Consistent with this possibility, Cardoso and colleagues [23] found that when monkeys closed their eyes, and arousal dropped, the hemodynamic signal in V1 rose slowly, although when performing a task the same signal increased when monkeys were more aroused.

Several studies have found that spatial attention lowers neural spiking variability [40, 57, 59–61]. Here, we have shown that arousal lowers neural response variability as well, albeit when measured with hemodynamics rather than spikes. However, we note a couple of distinctions between these sets of findings. First, we observed a decrease in variability in ipsilateral visual cortex, a subregion of visual cortex where no stimulus ever appeared and where participants were not attending. Second, studies on attention usually observe a decrease in ongoing neural fluctuations. The variability time series (Fig 7) combined with the simulation (Fig 6)

suggest that the variability decrease we see is not a result of diminished ongoing fluctuations but rather is associated with neural activity linked to the trial onset (i.e., the task-related response), and specifically the temporal properties of that activity. The similarities between arousal and attention raise the question of whether they are carried out through similar or distinct neural mechanisms.

Studies combining optical imaging with electrode recordings have found a task-related hemodynamic signal in monkey visual cortex that was not a response to visual stimulation and that did not correspond to neural spiking activity [1, 22, 23]. The signal they measured was entrained to task timing and was modulated by reward. They further found that reward increases response amplitude and lowers temporal variability, consistent with our results [23]. However, they found no difference in variability of response amplitude (Das, personal communication). Interestingly, when we excluded voxels that responded to the stimulus localizer, all effects remained significant except for the drop in amplitude variability with reward. We interpret this to indicate that this particular effect is less robust and requires the full population of responses in EVC to reach significance.

Our results agree with the optical imaging findings in several respects, suggesting that the task-related response measured in both species is one and the same. Sirotin and Das [1] did not find a clear correspondence between the task-related response and either neural spiking or LFP activity [62–64]. Consequentially, we cannot know whether the task-related response we have measured in humans reflects neural spiking or LFP. It may alternately reflect dynamics of spiking synchrony, intracellular subthreshold membrane voltage fluctuations, or non-neural sources of hemodynamic activity. Similarly, the changes in variability that we measured may not correspond to changes in neuronal spike rate variability but may rather correspond to other sources of variability, such as correlated neural variability, neural synchrony, or subthreshold membrane potential variability.

## Effect of arousal on stimulus-evoked activity

Recent studies in mice have characterized effects of arousal on stimulus responses at various stages of the visual system, including V1, lateral geniculate nucleus, superior colliculus, and retina [65–68]. Few studies have investigated the impact of arousal on visual responses in humans [69–71]. Our study was designed to minimize the impact of the visual stimulus. By using a small, brief stimulus, there was only very limited visually driven activity detected in the contralateral hemisphere in the region corresponding retinotopically to the location of the stimulus. As a result, it was not possible to characterize the impact of arousal on stimulus-driven responses. However, our findings have important implications for studying effects of arousal on stimulus responses. Since visual responses are usually obtained during a task, visual cortex will exhibit widespread task-related activity which is modulated by arousal. Therefore, before testing whether arousal impacts visual activity, it is necessary to tease apart and dissociate arousal effects on task-related activity from arousal effects on stimulus-evoked responses. Since both arousal and feedforward stimulus drive are present in visual cortex, distinguishing between these two sources of activity may not be trivial.

## Relationship between physiological processes and fMRI signal

The fMRI signal has been shown to covary with several physiological measures, including pupil size, heart rate, and respiration volume. Furthermore, physiological processes have been shown to be modulated by reward. This raises two questions. First, is task-related activity a result of a physiological process entraining to the task, rather than an endogenous neural response? Second, is the modulation of task-related activity with reward a secondary impact of

reward on physiological processes, rather than a direct link between reward and neural processes? We believe that both task-related activity and the modulation of this activity with reward are primarily neurogenic and not secondary effects of physiological changes in the periphery. Regarding the former question, if task-related activity was the result of physiological processes entraining to the task timing, we would expect to observe task-related activity that was primarily unitary throughout the brain, with relatively minor differences in latency related to blood vessel size and consequent differences in transit time of oxygenated blood. The stark difference in response timing that we observed between foveal and peripheral visual cortex rules out this possibility. To answer the latter question adequately, we would need to obtain measurements of many physiological processes, including measures such as skin conductance [72], metabolism [73, 74], hydration [75], and circadian phase [76, 77], regress them out of the fMRI data, and then test whether reward modulation is evident in the residual time series. Doing so is beyond the scope of this study. Instead, we regressed out the global signal, and while we observed a dramatic decrease in the covariation of the fMRI data with heart rate, we saw the same effects of reward that we observed in the original, raw data. Therefore, we think it unlikely that response modulation with reward is solely a result of changes in peripheral physiological processes.

## Heterogeneous task-related response phase

We consistently observed an inverted task-related response in and around the foveal representation of EVC, relative to the peripheral representation (see Figs 1,3 and 4). There are several possible interpretations for this inversion. First, the inversion may reflect changes in microsaccade rate. Microsaccade rate has been shown to decrease following stimulus presentation, followed by an increase around 400 ms after stimulus onset [78–81]. Microsaccades have also been shown to evoke BOLD responses in EVC [82]. The increase in microsaccade rate may result in increased BOLD activity at the fovea, where retinal slip of the fixation cross would be maximal. However, the increase in microsaccade rate would have to occur well after the end of the trial in order explain why the activity at the fovea is several seconds delayed relative to the task-related activity in the periphery, which seems unlikely given the oft-characterized time course of microsaccade rate modulation during a range of cognitive tasks [78–81].

An alternative explanation is that foveal activity reflects other processes unique to the fovea. For example, foveal EVC receives feedback from peripheral object-selective cortex [83]. In our experiment, stimuli were gratings, not objects, but foveal EVC may also receive feedback from peripheral extrastriate cortex. Feedback may result in later activity in foveal EVC, resulting in a later response in the fovea. However, we find it unlikely that feedback would take seconds to reach the fovea.

Yet a third possibility is that activity measured at the fovea is the result of blood stealing from more peripheral areas [84, 85]. Negative activity surrounding positive stimulus-evoked activity is often attributed either to a decrease in neural activity associated with surround suppression, or to blood stealing. However, in our case, there is no stimulus, ruling out surround suppression. However, blood stealing at the fovea is unlikely, since blood stealing would predict a negative activity on all sides of the positive task-related response, and we do not see negative activity in regions anterior to the positive activity. Moreover, blood stealing as an explanation for negative BOLD has been met with increasing skepticism [86, 87].

Finally, the inverted response at the fovea may reflect spatial attention focusing at fixation during the intertrial interval, followed by disengagement from fixation when the stimulus appears. Attention in between trials would cause an increase in the BOLD response at the fovea corresponding temporally to the task-related response observed there. However, we

observed a similar inversion of the response at the fovea during a task with no peripheral stimulus, where participants had no incentive to shift attention to the periphery (S1 Fig). This suggests that the peripheral stimulus and attention to it are not the cause of the inversion of the foveal task-related response. Nevertheless, it is possible that even without a peripheral stimulus the periodic task at fixation evoked periodic foveal attention that differed in timing from the task-related response, causing the inverted response at the fovea.

### Potential non-neural sources of task-related activity

In this study, we strove to minimize the chance of any head-motion or respiration effects in the data. First, we acquired data using a multiple-echo pulse sequence and preprocessed the data with ME-ICA denoising, which has been shown to greatly reduce effects of head movement, which are not BOLD-like in terms of TE dependence, and which generally manifest as focal changes [4]. However, ME-ICA does not fully remove effects of respiration, which generally consist of spatially widespread $T2^*$ signals. Although we obtained respiration traces concurrent with fMRI, Power and colleagues have shown that existing algorithms for modeling and removing respiratory signals are inadequate [35]. Therefore, we instead regressed out the global signal [11], an approach that has been shown to minimize respiratory effects [4, 38, 42, 88]. We can thus rule out both head motion and respiration as non-neural sources of the task-related response. We cannot, however, determine definitively that the task-related response is neurogenic. There may be other non-neural signals, such as vasomotion [6, 89], that give rise to the task-related response and/or account for the modulation of the response with reward.

### Conclusions

EVC is most often studied in relation to the processing of exogenous visual stimuli and the modulation of activity with a range of cognitive processes, such as attention, memory, and perceptual learning. Here, we have isolated an endogenously driven hemodynamic response and shown that it is related to participants' engagement in a task, as indexed by measures of arousal. Several important unresolved questions remain. For example, does this task-related response impact the processing of visual stimuli, akin to the modulation of visual responses with spatial attention [90]? What is the relationship between task-related activity and behavior [55]? And finally, optical imaging studies in monkeys have raised questions about the relationship between task-related hemodynamic activity and changes in spiking activity [1]. Do the changes that we observed in fMRI correspond to a change in electrophysiological measurements? We also note that there has been considerable, ongoing debate around global signal regression, particularly in the context of functional connectivity analysis of resting-state data [41, 91–93]. Although we take no side in that debate, we point out that the task-related response may constitute an important component of the global signal. Hence, our results suggest that a very large number of fMRI studies may have removed from the data an important component of the brain's response related to participants' engagement. Future studies will need to evaluate what role this endogenous hemodynamic component plays in cognitive processes and through which underlying neural computations it is accomplished.

## Methods

### Stimulus and task

Participants were instructed to continuously fixate a small (0.7 deg) central cross while performing a peripheral two-alternative forced choice (2AFC) orientation discrimination task (Fig 1A). Each trial lasted for 15 s. The trial began with the appearance of a small oriented

grating for 200 ms. Participants determined whether the grating was tilted clockwise or counterclockwise relative to vertical and responded with a button press. The interstimulus interval was 14.8 s. Stimuli were generated using Matlab (MathWorks, MA) and MGL [94] on a Macintosh computer.

The stimulus consisted of a full contrast grating with a spatial frequency of 4 cycles/degree (cpd) windowed by a 1-deg-diameter circular aperture (raised cosine with a 0.25-deg transition zone). The stimulus was presented at a 5-deg eccentricity, in the right visual field, 45˚ below the horizontal meridian. Participants responded with a button press and immediately received auditory feedback indicating whether or not they were correct. Participants performed this task under one of two reward conditions that differed in the amount of potential monetary gains and losses. Participants were instructed prior to each run whether this would be a high- or low-reward run. They were informed of their gains and losses (both for that run, and cumulatively for the session) at the end of each run.

## fMRI experiment

Fifteen participants (5 male, 12 right-handed) participated in the fMRI experiment. Task difficulty was controlled by manipulating the tilt angle of the stimulus away from vertical. Tilt angle was determined in a separate run using a staircase procedure (1 up, 2 down staircase, with 0.1˚ increments in tilt angle) in order to roughly equate difficulty across participants (tilt mean ± std: 1.25˚ ± 0.44˚). Participants gained monetary reward for every correct response and lost money for every incorrect response.

Each run was either a high-reward run or a low-reward run and consisted of 16 trials. There were two versions of the experiment that differed in terms of the maximal reward per run, and whether the rewarded amount was determined based on performance on the entire run, or for a single randomly chosen trial. In one version, participants were rewarded for every correct trial and could gain up to $16.16 per high-reward run and up to $0.16 per low-reward run ($N = 14$ participants). In the second version, participants were rewarded for a single randomly chosen trial, a sum of $20 for a high-reward run and $0.25 for a low-reward run ($N = 6$ participants). Before each run, participants were notified whether the upcoming run was a high- or low-reward run and how much money they could gain. At the end of each run, participants were informed how much money they had gained during that run. Results were similar across the two versions of the experiment, and therefore, data were combined across versions. For five participants who participated in two sessions, data were concatenated across sessions. Thus, we collected data in a total of 20 sessions across 15 participants. Each run consisted of 16 trials, lasting a total of 160 volumes, or 240 s. Each participant completed 10–16 runs per session.

## Ethics statement

Participants provided written informed consent. The consent and experimental protocol were in compliance with the safety guidelines for MRI research and were approved by the Institutional Review Board at National Institutes of Health (protocol number 93-M-0170).

## Stimulus localizer

In addition to the main experiment, each scanning session included a stimulus localizer run, consisting of alternating 9-s blocks of right and left visual field stimulation. During each block, the identical stimulus that appeared during the main experiment appeared either in the same location as in the main experiment (5-deg eccentricity, in the right visual field, 45˚ below the horizontal meridian) or in the mirror symmetric location in the left visual field. During each

9-s block, the grating changed orientation and phase every 200 ms in order to avoid adaptation. Each localizer run lasted 168 fMRI volumes (14 cycles of the stimulus).

## Stimulus-free experiment

Participants were instructed to continuously fixate a small central dot on a black background. Every 15 s, the dot luminance decreased slightly for 100 ms, and participants were instructed to respond with a button press. Each run lasted 164 volumes. Participants ($n = 6$) typically completed 10 runs of this experiment in a single session.

## fMRI scanning

MRI scanning was carried out on a research-dedicated GE 3T Sigma scanner, using a 32-channel head coil, at the Functional Magnetic Imaging Core Facility at NIH. Functional scans were acquired using T2*-weighted, gradient recalled echo-planar imaging to measure BOLD changes in image intensity [95]. Functional imaging was conducted with 22 slices oriented perpendicular to the calcarine sulcus and positioned with the most posterior slice at the occipital pole, covering all occipital and posterior parietal and temporal cortex (TR: 1,500 ms; multi-echo TEs: 14.2, 30.1, and 46 ms; FA: 75˚; voxel size: $3 \times 3 \times 3$ mm; grid size: $64 \times 64$ voxels). For each participant and in each session, a high-resolution T1-wieghted anatomy of the entire brain was acquired (magnetization-prepared rapid-acquisition gradient echo [MPRAGE]; TR: 2,500 ms; TE: 3.48 ms; FA: 7˚; voxel size: $1 \times 1 \times 1$ mm; grid size: $256 \times 256$ voxels; 172 slices). The anatomical volume was used for co-registration across scanning sessions and for gray matter segmentation and cortical flattening.

## Physiological monitoring

Heart rate was monitored using a pulse oximeter at 50 Hz. The reciprocal of intervals between peaks defined the instantaneous heart rate which was then linearly interpolated. Pulse-to-BOLD kernel was obtained by performing regression of the concatenated fMRI time series with heart rate, separately for high and low reward. Kernels were then averaged across reward and participants. Amplitude of heart rate and kernels were calculated by computing the std. One participant was excluded from analysis of heart rate because of poor-quality measurements.

## Exclusion criteria

One participant was excluded from the analysis because of a response pattern that was more than 2.5 std from the group of participants. The main results were similar, albeit slightly less robust, when this participant was included in the analysis.

## fMRI data analysis: Preprocessing

Data were motion corrected (linear interpolation) using AFNI software using images from the first echo (14.2 ms), which had the highest gray matter/white matter contrast. Time-series data from the three echo times were then combined using an ICA-based denoising procedure (ME-ICA) implemented in Python (meica.py) [33]. Time-series data were subsequently processed with mrTools [96] and custom Matlab functions.

## Regions of interest

Boundaries of visual areas V1, V2, and V3 and eccentricity maps were applied from an anatomical template of retinotopy [97]. Ipsilateral V1, V2, and V3 were combined to create a

single EVC ROI. We further divided EVC into 12 eccentricity bins [98] in which bin width increased exponentially with eccentricity from 0.2 to 70 deg, resulting in bins with roughly equal numbers of voxels. We analyzed the task-related response separately within each of the bins and averaged the response amplitudes across participants.

## fMRI data analysis: Main experiment

Data from the first trial (10 frames) at the beginning of each functional run were discarded to minimize the effect of transient magnetic saturation and to allow hemodynamic response to reach steady state, resulting in a time series of 15 cycles' length (150 frames). The time series for each voxel was divided by its mean to convert from arbitrary intensity units to percent change in image intensity. Subsequently, the time series of each run was converted to z-score values, by dividing by the std. When analyses were performed without z-scoring, all effects remained significant (S4 and S5 Figs). Finally, to remove physiological signals from the data, the mean signal across the entire scanned volume was regressed out from each voxel's time series [4]. Time series belonging to all voxels within each ROI were averaged together.

For correlation analysis, all runs were averaged regardless of reward type, and each individual voxel's time course was fitted to a cosine. Each voxel was then assigned the correlation coefficient and phase of the best-fitting cosine [99]. For all other analyses, all runs within each reward type (high or low) were concatenated. Next, all voxels' time series were averaged together to yield a single time series. To quantify the task-related response amplitude, all trials were averaged to yield a single average trial time series, consisting of 10 time points. Trials in which the participant did not respond were not analyzed.

Response amplitude was measured by computing the std of the mean trial time series. To estimate response latency, we computed the Fourier transform of the trial-locked average and took the phase of the second component, corresponding to the phase of the signal at the trial frequency. A second measure of response amplitude consisted of computing the amplitude of the second component of the Fourier transformed mean trial.

To test for a difference in response amplitude between high- and low-reward runs, a non-parametric permutation test was used. For each participant, we permuted the reward labels (high or low) for each trial, before averaging across trials. We then recomputed the std and subtracted the low-reward amplitude from the high-reward amplitude. These differences were then averaged across participants to get a group average. We repeated this 10,000 times to get a null distribution of mean amplitude differences. Finally, we evaluated the actual mean difference to between high- and low-reward amplitudes against this null distribution. The *p*-value is the fraction of permutations that resulted in equal or higher differences than the actual difference.

For the Fourier analysis, after averaging trials within reward type, and averaging across voxels, we computed the absolute value of the Fourier transform of the average trial. Averaging the spectrum across participants yielded a group average Fourier spectrum. To test for a difference in Fourier amplitude and latency between high- and low-reward runs, we used the same nonparametric permutation procedure described above for response amplitude.

Response variability was measured in three ways. First, we divided all time series into single trials. The std of each time point was computed and then averaged across the 10 time points, yielding the mean time-point variability. Next, we performed Fourier transform on each individual trial and extracted the phase of the response at the trial frequency. Temporal variability was defined as the circular std of the phase. Finally, as a third measure of response variability, we computed the std of each trial, yielding an amplitude estimate per trial. Amplitude variability was defined as the std of that amplitude. All three measures were averaged across

participants. Permutation test procedure was identical to that performed for response amplitude and latency.

## Retinotopic maps

For six participants, retinotopy was measured in a separate session, at 7T, using nonperiodic traveling bar stimuli and analyzed using the population receptive field (pRF) method [100]. Bars were 3 deg wide and traversed the field of view in sweeps lasting 24 s. Eight different bar configurations (four orientations and two traversal directions) were presented. Data were acquired on a research-dedicated Siemens 7T Magnetom scanner using a 32-channel head coil. Functional imaging was conducted with 54 slices oriented perpendicular to the calcarine sulcus covering the posterior half of the brain (TR: 1,500 ms; TE: 23 ms; FA: 55°; voxel size: $1.2 \times 1.2 \times 1.2$ mm with 10% gap between slices, respectively; grid size: $160 \times 160$ voxels. Multi-band factor 2, GRAPPA/iPAT factor 3). The pRF of each voxel was estimated using standard fitting procedures [100], implemented in Matlab using mrTools. A map of the eccentricity of the center of each voxel's fitted pRF was used solely for visual comparison to maps of task-related activity.

## Stimulus-free data analysis

The first 14 volumes were discarded, leaving 150 volumes, or 15 cycles. Runs were z-scored and concatenated. For correlation analysis, all runs were averaged, and each individual voxel's time course was fitted to a cosine. Each voxel was then assigned the correlation coefficient and phase of the best-fitting cosine. For analysis of response amplitude, all runs were concatenated, all voxels' time series within each eccentricity bin were averaged together, and all trials were averaged to yield a single mean trial. Response amplitude was measured by computing the std of the mean trial.

## Localizer data analysis

The first cycle was discarded, leaving 13 cycles. Each individual voxel's time course was fitted to a cosine with a period matching the cycle duration of 12 volumes (18 s). Each voxel was then assigned the correlation coefficient and phase of the best-fitting cosine. For participants who participated in two sessions, the two localizer runs were concatenated. Each voxel was then assigned the correlation coefficient and phase of the best-fitting cosine.

## Visual field plots

We used an inverted encoding model [101, 102] to project activity patterns into visual space. The phase of the best-fitting cosine, *ph*, and the coherence between that cosine and the voxel's time series, *co*, were converted to a complex response: $c = co \times (\cos ph + i \sin ph)$. We obtained pRF center estimates from an anatomical template [97]. The template does not provide pRF size estimates, so we assumed that pRF size increases approximately linearly with eccentricity and increases along the visual hierarchy [100, 103, 104]. We therefore modeled pRF size as a function of pRF center eccentricity and cortical region. Each voxel's pRF was modeled as a gaussian with size: $\sigma = 0.2 \times r \times roi^{0.7}$, were *r* is the pRF center eccentricity, and *roi* was 1, 2, or 3 for V1, V2, and V3, respectively. Eccentricity was assigned according to an anatomical template [97]. Finally, for each pixel in the visual field, complex responses were summed across all voxels, weighted by their gaussian pRF. The phase of the mean complex responses was plotted across the visual field, with opacity scaled by the coherence of the mean complex responses.

## Simulation

The simulation consisted of 100 runs per noise level. Each run consisted of 16 trials, of 10 time points each. The task-related response was a double gamma IRF convolved with the first time point of each trial. Temporal noise determined the std of gaussian noise added to the timing of each trial. To implement amplitude noise, a sigmoidal function was applied to gaussian noise multiplied by the noise amplitude and added to each trial's response amplitude. Finally, ongoing fluctuations were modeled as 1/f noise that was added to the signal. The first trial of each run was removed, and the remaining time series underwent high-pass filtering. All runs for a given noise level were concatenated and separated into trials. We then computed the mean trial and its std, which is the time-point variability. We repeated this procedure with two different IRFs.

## Pupil-size measurements

Participants performed the 2AFC orientation task (described above) while their eye position and pupil size were recorded (tilt mean ± std: 1.13˚ ± 0.58˚). Each individual participated in one of two versions, which differed both in terms of the stimulus onset asynchrony (SOA) and reward protocol. In one version, the SOA was randomly chosen on each trial to be 4, 6, or 8 s. In this version, maximal rewards were $16.80 on high-reward runs and $0.42 on low-reward runs, according to the number of correct trials ($N$ = 10 participants). In the second version of the experiment, the SOA was fixed at 6 s, and the maximal reward per run was $10 for high-reward runs and $0.01 for low-reward runs. For this version of the experiment, the actual reward delivered was determined by performance on a randomly selected single trial ($N$ = 3 participants). Eye position and pupil size were recorded with Eyelink 1000 Plus, at a temporal sampling resolution of 500 Hz. We measured pupil size both concurrent with fMRI scanning and in the psychophysics lab; the latter measures were considerably higher quality and are reported here. Eleven of the individuals who participated in the fMRI experiment took part in the additional pupil-size measurements.

## Eye data analysis

Blinks were removed from the data, including three time points before and three time points after each blink. Next, data were segmented into trials. To combine data from all experiment versions, we analyzed only the first 4,000 ms of each trial. Trials in which the participant did not respond were not analyzed. Tonic pupil size was quantified by measuring the mean pupil size during the first 50 ms of every trial and averaging across trials. Phasic changes in pupil size were quantified by measuring the std of each trial's time series. To test whether phasic and tonic pupil sizes were larger for high-reward runs, a nonparametric permutation test was used. For each participant, we permuted the reward labels (high or low) for each trial, computed the mean phasic and mean tonic pupil sizes for both reward conditions, and subtracted the value for low-reward from the high-reward value. These phasic and tonic differences were then averaged across participants, generating a group average. We repeated this procedure 10,000 times to generate a null distribution of mean phasic and tonic high-reward minus low-reward differences under the null hypothesis that there was no difference between high and low reward. We then evaluated the actual phasic and tonic differences against this null distribution. The $p$-value equals the fraction of permutations that resulted in equal or higher differences than the actual difference.

## Supporting information

**S1 Fig. Task-related activity in visual cortex in the absence of visual stimulus.** (A) Medial view (inset) and a flattened map of right hemisphere visual cortex (bottom panels) of

participant P16. Left: visual eccentricity. Hue indicates preferred eccentricity for each voxel. Retinotopic borders of V1–V3 were defined by an anatomical template extending to 80˚ eccentricity, well beyond the spatial extent of the screen. Map threshold, r > 0.3. Shaded region on lateral and medial views indicates cortex not included in the imaged/field of view. Center: response correlation for control experiment, showing a widespread fMRI response linked to task timing. Map threshold, r > 0.3. Hue indicates correlation with best-fitting cosine at the task frequency. Right: response phase. Same threshold as middle panel, with hue indicating phase of best-fitting cosine for each voxel. Phase values indicate the response latency for each voxel. (B) EVC task-related response amplitude as function of eccentricity. Shaded regions, ±SEM across participants. Amplitude varies with eccentricity in a similar manner to the main experiment; compare with Fig 4A. EVC, early visual cortex; fMRI, functional MRI.
(TIF)

**S2 Fig. Task-related activity is distinct from visual activity.** Phase and coherence of visual cortex responses retinotopically projected back to the visual field, averaged across *n* = 14 participants. Opacity of each pixel reflects the coherence of the time series obtained by averaging across voxels, weighted by their retinotopic response to that pixel. Hue reflects the phase of the resulting time series. Left, average of localizer runs. Center, average of high-reward runs. Right, average of low-reward runs. Voxels are from combined right- and left-hemisphere EVC ROIs. Localizer runs evoked localized activity limited to voxels with pRFs that overlap the stimulus. In contrast, task runs evoked widespread activity that did not correspond retinotopically to the stimulus. The inverted response is not visible in high- and low-reward runs because of the small size of foveal pRFs. Foveal voxels have small pRF sizes, overlapping with more peripheral pRFs at different phases. This results in low coherence at the fovea, i.e., a small opaque area at the center. EVC, early visual cortex; pRF, population receptive field; ROI, region of interest.
(TIF)

**S3 Fig. Saccades during pupil-size experiment.** Saccades occurring within the first 1,000 ms of each trial were analyzed. (A) Main sequence, pooling saccades across participants, demonstrates a linear relationship in log-log axes between peak velocity and saccade amplitude, as expected by the biomechanics of the oculomotor plant. (B) Amplitude distribution of saccades. Most saccades were small (<1 deg) and are hence considered microsaccades. Median saccade amplitude, 0.52 deg. (C) Spatial distribution of saccades. Each dot represents the displacement of a saccade relative to the origin (0,0). (D) Direction distribution of saccades. Saccades are generally horizontal and were not directed toward the target, nor were they of sufficient amplitude to reach the target.
(TIF)

**S4 Fig. Effect of reward on response amplitude without fMRI time-series z-scoring.** For this supplementary analysis, time series were not z-scored. In all other respects, analysis and figure are identical to Fig 4. High reward had significantly higher amplitude (*p* = 0.004) and lower time-point variability (*p* = 0.0012), temporal variability (*p* = 0.0218), and amplitude variability (*p* = 0.0423). fMRI, functional MRI.
(TIF)

**S5 Fig. Effect of reward on response variability without fMRI time-series z-scoring.** For this analysis, time series were not z-scored. In all other respects, analysis and figure are identical to Fig 5. High reward had significantly lower time-point variability (*p* = 0.0012), temporal variability (*p* = 0.0218), and amplitude variability (*p* = 0.0423) than low reward. Time-point variability was significantly greater for low reward in each of the eccentricity bins (*p* < 0.01 for

all). fMRI, functional MRI.
(TIF)

**S6 Fig. Simulated response amplitude as function of noise level for first (left panel) and second (right panel) simulations.** Amplitude was computed as the standard deviation of the time series presented in Fig 6 and normalized relative to amplitude at noise level 1 (i.e., no noise). Amount of independent noise and amplitude jitter had no systematic impact on response amplitude, whereas amplitude drops monotonically with increasing temporal jitter.
(TIF)

# Acknowledgments

Special thanks to Aniruddha Das, Chris Baker, David Heeger, Charlie Burlingham, and members of the Laboratory of Brain and Cognition at NIMH for constructive discussions and helpful comments and to Tenzin Yin for help with data analysis.

# Author Contributions

**Conceptualization:** Elisha P. Merriam.

**Data curation:** Zvi N. Roth.

**Formal analysis:** Zvi N. Roth, Minyoung Ryoo, Elisha P. Merriam.

**Funding acquisition:** Elisha P. Merriam.

**Investigation:** Zvi N. Roth, Minyoung Ryoo, Elisha P. Merriam.

**Methodology:** Zvi N. Roth, Elisha P. Merriam.

**Project administration:** Elisha P. Merriam.

**Resources:** Elisha P. Merriam.

**Software:** Zvi N. Roth, Elisha P. Merriam.

**Supervision:** Elisha P. Merriam.

**Visualization:** Zvi N. Roth, Elisha P. Merriam.

**Writing – original draft:** Zvi N. Roth, Elisha P. Merriam.

**Writing – review & editing:** Zvi N. Roth, Elisha P. Merriam.

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
