## [Editor Report · Decision Letter 0]

11 Mar 2020

Dear Dr Roth, 

Thank you for submitting your manuscript entitled "Task-related activity in human visual cortex" for consideration as a Research Article by PLOS Biology.

Your manuscript has now been evaluated by the PLOS Biology editorial staff, as well as by an Academic Editor with relevant expertise, and I am writing to let you know that we would like to send your submission out for external peer review. Please accept my apologies for the delay in sending this initial decision to you.

Before we can send your manuscript to reviewers, we need you to complete your submission by providing the metadata that is required for full assessment. To this end, please login to Editorial Manager where you will find the paper in the 'Submissions Needing Revisions' folder on your homepage. Please click 'Revise Submission' from the Action Links and complete all additional questions in the submission questionnaire.

Please re-submit your manuscript within two working days, i.e. by Mar 13 2020 11:59PM.

Kind regards,

Gabriel Gasque, Ph.D.,

Senior Editor

PLOS Biology

---

## [Decision Letter · Decision Letter 1]

5 Jun 2020

Dear Dr Roth,

Thank you very much for submitting your manuscript "Task-related activity in human visual cortex" for consideration as a Research Article at PLOS Biology. Your manuscript has been evaluated by the PLOS Biology editors, by an Academic Editor with relevant expertise, and by two independent reviewers (Reviewer 1, Kristina Visscher, has signed her comments). Please accept my sincere apologies for the very long delay in communicating this decision to you. 

In light of the reviews (below), we will not be able to accept the current version of the manuscript, but we would welcome re-submission of a much-revised version that takes into account the reviewers' comments. We cannot make any decision about publication until we have seen the revised manuscript and your response to the reviewers' comments. Your revised manuscript is also likely to be sent for further evaluation by the reviewers.

We expect to receive your revised manuscript within 2 months. 

**IMPORTANT - SUBMITTING YOUR REVISION**

Your revisions should address the specific points made by each reviewer. Having discussed these comments with the Academic Editor, four things are more salient to us and should be addressed thoroughly:

1) The claim of no stimulus driven effect by an ipsilateral stimulus. We know that feature-selective modulations can spread to the ipsilateral regions, so this assumption needs careful justification. 

2) The issues about variability in each condition and the dependent measure of the BOLD response also need careful consideration. 

3) Reviewer 2 expressed important qualifications that need to be included about valance/arousal

4) Reviewer 2 also wondered about stimulus driven responses - this seems fairly important to cover more thoroughly. 

Please submit the following files along with your revised manuscript:

*Re-submission Checklist*

*Published Peer Review*

*PLOS Data Policy*

*Blot and Gel Data Policy*

Sincerely,

Gabriel Gasque, Ph.D., 

Senior Editor

PLOS Biology

REVIEWS:

Reviewer #1, Kristina Visscher: Summary: 

Roth et al., investigate the relationship between arousal and task-related neural activity in visual cortex. The authors measure fMRI signals in V1 during an orientation discrimination task, while simultaneously recording pupil size and heart rate as measures of arousal. 

They tested the hypothesis that task-related evidence tracks arousal. 

First they show that pupil size and heart rate change during trials of the task. They show also that global mean regression removes the signal associated with pulse. A selective averaging type approach was used to measure the level of fMRI activity in response to trials, in units of standard deviation of BOLD signal across the run duration. Trial by trial variability in timepoint, amplitude and time to peak were smaller in the high than the low reward case. 

Another interesting result was that the amplitude (and, to a lesser degree, variability) of MRI response was very small at brain voxels corresponding to ~2-4 degrees eccentricity, but was larger at smaller and at larger eccentricities. This 'dip' is on its face unexpected, and the authors give a number of possible interpretations of these data in the discussion section. The authors also perform a simulation to show how trial to trial variability in various aspects of neural activity could lead to changes in the standard deviation of a signal.

The major inferences from these results are that:

1) Because the response follows dynamics similar to that of pupil size and heart rate, task -related activity is related to arousal

a. I think this isn't a very strong argument; granted, the authors do hedge, and use the term "suggest." However, it seems reasonable from first principles (and because the pupil size and heart rate change with reward condition) that the high reward and low reward conditions could differ in arousal. That's the only argument I think is needed to make later arguments about arousal being a likely factor contributing to the observed effects.

2) Higher reward increased response amplitude (measured in Fig 3) and decreased trial to trial variability (Figure 5, right side), suggesting that arousal (reward task) decreases trial-to-trial variability in responses.

3) There is a very interesting and hard to interpret result shown in Figure 4, where response amplitude depends in an interesting way on eccentricity.

Major:

* The authors assert that the changes in neural activity observed during the orientation discrimination task was due to arousal. However, it seems that there may be reason to believe that the changes are likely due to other top-down influences outside of arousal. The logic of the paper rests on this idea, e.g., from line 333 in discussion "This activity is not evoked by the stimulus, since there was never a stimulus contralateral to the hemisphere we analyzed." 

There needs to be stronger evidence that activity ipsilateral to a stimulus is due to arousal and not, e.g., activity driven by the strong functional coupling between mirror symmetric portions of cortex, including V1. If there was activity in response to the stimulus in the contralateral side, it seems reasonable to suppose that there would be activity in response to that activity in the highly functionally correlated opposite side. The authors should provide description of why the argument that activity in ipsilateral visual cortex could not come from ipsilateral stimulation. [of course, ipsilateral activity would be far less than contralateral activity. But for the connectivity reasons given above, and due to small effects in other datasets, it seems reasonable to suppose that some activity in ipsilateral cortex could be predicted.]

* Response amplitude is measured in std. Why is this choice of units used? It seems to be the STD across the whole run. Thus, on a run with higher variability due to task (like the high reward condition), we would expect the total STD to be larger. So the excursion due to any given trial would be artificially smaller in that case. I think it warrants giving a clear justification of why STD was used instead of % BOLD change, or another measure, especially because (next bullet point), it appears that this choice could result in false positives.

In particular, why do MRI responses not start at time 0 with value = 0 (fig 3, they all seem to be significantly different from 0).

* Because there is a smaller amplitude of signal in the low reward case, the STD units in that case would correspond to smaller numbers in units of % change in BOLD (for example). Thus, the same change in BOLD would correspond to a larger STD value for the low reward case than the high reward case. Thus, the same variability in BOLD would be a larger in low reqard than high reward case. Can you show that this simplistic explanation does not explain the data from Figure 5? One way to do this would be to re-calculate the values in terms of % change from baseline (where baseline is defined as activity at time 0, not as mean across all signals -- standard GLM analysis would work). There may be other logical ways to get around this issue, but the bigger picture is: how do we convince ourselves that the effect in Fig 5 is not due to the choice of units.

* One of the major interpretations from the data seems to be that trial to trial variability decreased with reward. And all the figures show response amplitude differences that depend on eccentricity (a lot). And yet, there are no plots showing how timepoint variability changes with eccentricity. This seems like it might lend support to the arguments, or at least be informative about the origins of this effect. The authors should add that figure, or explain why it is not included.

Minor: 

* fits to the models needed to measure amplitude 

* I assume trials were presented one per 15 seconds (based on doing some division based on lines 579 to 580). Did I miss this point? Was a GLM used for data analysis? 

* How many trials per condition were included? (run is 16 trials, there were 20 sessions. But how many runs per session?)

* Should line 578-579 read "total of 20 sessions in each of 15 subjects"?

* In line 322, "the difference between the two exhibits a temporal profile similar to each of the two reward conditions" -- it's unclear (at least to this reader) what the temporal profile being described is. It implies that Figure 7B is similar to another figure, but it's unclear which figure? (is it 3B? I don't see the similarity)

* In line 352, it is noted that the modulation of task related responses with reward resembled modulation of pupil size. Please indicate which data are used for that conclusion. If the argument is that pupil size effects are larger in high than low arousal conditions, and fmri effects are larger in high than low arousal conditions, this isn't compelling evidence that "task-related activity is related to arousal." 

* Around line 474, the text refers to the shape of the effect exemplified in Figure 4a as "an inverted task-related response in and around the foveal representation" -- I find this a confusing characterization, since the very small eccentricities (around 0) have strongest responses, while the areas outside (there isn't an axis label, but it looks like around 4 degrees -- which is technically outside the fovea) have the weakest responses. Why not call it "a dip in response around eccentricity of 4 degrees" or something like that. The characterization as it stands suggests that there is no response at fovea, when the largest response is at fovea.

* Line 484 refers to the "oft-characterized time course of microsaccade rate modulation during a range of cognitive tasks" -- citation please. 

* The section about "heterogeneous task-related response phase" gives good arguments against each of the described explanations, except the last one. It would be worth more detail on the last item if it is the most likely possible explanation.

* In line 60, the authors state that their hypothesis is that "task-related evidence tracks arousal." The use of the word "evidence" seems an odd word choice. It may be worth changing this statement to task-related responses tracks arousal

* In line 79 , since this appears to be the first use of the abbreviation "CW" and "CCW", they should be written out as clockwise (CW) and counter clockwise (CCW).

* Line 86 reads Shaded region indicates cortex not included in field of view." -- it's unclear to this region what shaded region refers to (on the flattened map? On the hemisphere?)

* To make it clear to readers who may not be familiar with units of spatial frequency, the authors should rewrite cpd as cylces/deg in line 549.

* Citations are for the methods sections are sparse. If based on techniques used in prior work, the authors should cite the relevant sources. 

* Was there any eye tracking data showing that subjects were fixating centrally? If not, it's important to show data from the hemisphere where you expected activity to make sure that the activation in located in the location of the stimulus.

Reviewer #2: Task-related activity in human visual cortex

In this manuscript, the authors investigate the role that task based activity plays in modulating visuocortcial responses. To do so, they manipulate the value of accuracy-based reward on a fine orientation discrimination task, and find modulation of pupil diameter as a function of reward, consistent with an increase arousal. Importantly, the authors also observed increases in BOLD response in early visual cortex, independent of regions of visual stimulation, and a decrease in trial-to-trial variability in BOLD response. The authors then propose a simple model wherein increased temporal precision can underlie their effects. The topic is interesting and timely, and the study is generally well designed and analyses carefully conducted. Below I outline my specific, albeit minor concerns.

1. Based on the pupil and heart rate measures, the authors conclude that their reward modulation was an effective driver of arousal. While in principle I agree, the authors should note that they have not fully disentangled reward from arousal win this study. A pure effect of arousal would be evident in a valence-free modulation of pupil diameter and heart rate (and presumably BOLD response in visual cortex). In other words, if the study was carried out with punishment rather than reward, the authors would expect the same modulatory effects, were they driven purely by arousal state. Now, I don't think these observed effects in this study are *not* arousal-driven, but the authors should acknowledge that in this design reward has not been dissociated from arousal.

2. There was a small, full contrast grating for which participants carried out an orientation discrimination task on. And yet, the manuscript largely neglected to focus on any reward0-related modulation of the stimulus-evoked BOLD response. I get the impression there was no functional localizer carried out, but given that there was pRF mapping for the participants, would it be possible to look how the representation of the visual stimulus was affected by the reward value manipulation? 

3. In line 55, authors state that "changes in arousal lead to pupil dilation and do not increase sensitivity [15, 20]." This is not necessarily true. There are plenty of behavioral studies that have demonstrated improvements in perceptual performance with arousal state, including cognitive performance and low level perceptual sensitivity.

4. Did the authors observe any correlation between the time series for pupil or heart rate, and BOLD modulation? In other words, within a condition (High or low reward), were there fluctuations in physiological measures that squared with fluctuations in magnitude of BOLD response.

---

## [Decision Letter · Decision Letter 2]

2 Sep 2020

Dear Dr Roth,

Thank you for submitting your revised Research Article entitled "Task-related activity in human visual cortex" for publication in PLOS Biology. I have now obtained advice from the original reviewers and have discussed their comments with the Academic Editor. Please accept my apologies for the delay in communicating this decision to you. You will note that both reviewers, Kristina Maria Visscher and Sam Ling, have identified themselves. 

We're delighted to let you know that we're now editorially satisfied with your manuscript. However before we can formally accept your paper and consider it "in press", we also need to ensure that your article conforms to our guidelines. A member of our team will be in touch shortly with a set of requests. As we can't proceed until these requirements are met, your swift response will help prevent delays to publication. Please also make sure to address the data and other policy-related requests noted at the end of this email.

*Copyediting*

*Published Peer Review History*

*Early Version*

*Submitting Your Revision*

Sincerely,

Gabriel Gasque, Ph.D.,

Senior Editor,

ggasque@plos.org,

PLOS Biology

ETHICS STATEMENT:

-- Please indicate within your manuscript if you experimental protocol adhered to the Declaration of Helsinki’s principles or any other specific national or international ethical guideline.

-- Please include the ID number of the protocol approved by the Institutional Review Board at National Institutes of Health.

DATA POLICY:

Note that we do not require all raw data. Rather, we ask for all individual quantitative observations that underlie the data summarized in the figures and results of your paper. For an example see here: http://www.plosbiology.org/article/info%3Adoi%2F10.1371%2Fjournal.pbio.1001908#s5

These data can be made available in one of the following forms:

Regardless of the method selected, please ensure that you provide the individual numerical values that underlie the summary data displayed in the following figure panels: Figures 2A-C, 3, 4A-C, 5A-C, 6A-H, 7AB, S1B, S2BD, S4A-C, and S5A-C.

Please also ensure that each figure legend in your manuscript include information on where the underlying data can be found, and ensure your supplemental data file/s has a legend.

Reviewer remarks:

Reviewer #1, Kristina Visscher: he authors made a very thorough response to my comments and, I felt, those of Reviewer 2. 

Reviewer #2, Sam Ling: The authors have done a wonderful job addressing my concerns, and I'm excited to see this work published!

---

## [Editor Report · Decision Letter 3]

21 Sep 2020

Dear Dr Roth,

On behalf of my colleagues and the Academic Editor, John Serences, I am pleased to inform you that we will be delighted to publish your Research Article in PLOS Biology. 

Early Version

PRESS 

Kind regards,

Alice Musson

Publishing Editor, 

PLOS Biology

on behalf of

Gabriel Gasque,

Senior Editor

PLOS Biology